

# Integrating network topology metrics into studies of catchment-level effects on habitat diversity

Eleanore L. Heasley[1], Nicholas J. Clifford[2], and James D. A. Millington[1]

[1] Department of Geography, King's College London, London, UK
[2] School of Social, Political and Geographical Sciences, Loughborough University, Leicestershire, UK

*Correspondence to:* Eleanore L. Heasley (eleanore.heasley@kcl.ac.uk)

**Abstract.** The spatial arrangement of the river network is a fundamental characteristic of the catchment, acting as a conduit between catchment-level effects and in-channel morphology and ecology. Yet river network structure is often simplified to reflect an up-to-downstream gradient of in-channel features, commonly represented by stream order. The aim of this study is to quantify network topological structure using new metrics – distance network density and elevation network density – that better account for the multi-dimensional nature of the catchment and which are functionally applicable across geomorphological, hydrological and ecological attributes of the catchment. The functional utility of the metrics in explaining patterns of physical habitat diversity is assessed in comparison to stream order. The metrics are calculated for four low-energy, anthropogenically modified catchments in the UK and compared to a physical habitat diversity score derived from England's River Habitat Survey. The results indicate that the new metrics offer a richer, and functionally more-relevant description of network topology than stream order, highlighting differences in the density and spatial arrangement of each catchment's internal network structure. Correlations between the new metrics and physical habitat diversity score show that distance network density is positively related to maximum habitat diversity in three of the four catchments. There is also evidence that increased distance network density may reduce minimum habitat diversity in catchments with greater anthropogenic modification. When all catchments are combined, distance network density is positively correlated with maximum, mean and minimum habitat diversity. There are no significant correlations between elevation network density and habitat diversity. In all but the largest streams, there is no significant relation between habitat diversity and stream order highlighting the limitations of stream order in accounting for network topology. Overall, the results suggest that distance network density is a more powerful metric which conceptually provides an improved method of accounting for the impacts of network topology on the fluvial system exhibiting strong relationships with habitat diversity, particularly maximum habitat diversity.

## 1 Introduction

Rivers are integrators of many elements of their catchments (Dovers and Day, 1988). Consequently, integrated catchment management has long been seen as the gold-standard for river management and has been adopted in catchments across the globe (Newson, 2008). Research linking patterns of in-channel features to catchment-level functioning is currently focussed on characteristics of the terrestrial catchment such as land cover, geology and topography (e.g. Cohen et al., 1998; Harvey et al., 2008; Jusik et al., 2015; Naura et al., 2016; Richards et al., 1996; Richards et al., 1997). Yet, 'hot-spots' of activity within catchments are identified based on the hydrological connectivity of the catchment (Newson, 2010), a characteristic that is often neglected by catchment-level studies. This missing component of the catchment is critical for true integrated catchment management as the impacts of key management features (e.g. water, channel, land, ecology and human activity)



40 are transmitted throughout the river network (Downs et al., 1991). By investigating the impacts of hydrological connectivity on river form and function, our understanding of catchment-functioning can become more holistic and beneficial to catchment management.

Effective catchment management rests not only on improving scientific understanding of river form and function across multiple scales, but also on better integration between the key disciplines of catchment studies:
45 geomorphology, hydrology and ecology. This type of interdisciplinary approach is critical for understanding complex multi-casual relationships in river systems (Dollar et al., 2007). However, catchment connectivity is parametrised differently by different disciplines based on their interests. The discipline of geomorphology focusses on characterising the morphometry of the catchment, either using general variables which are continuous across the landscape (e.g. elevation, slope, curvature etc.) or specific variables which represent
50 individual features such as catchments (e.g. drainage density, shape, area) or streams (e.g. stream order, stream length) (Evans and Minár, 2011). Hydrology focusses on how the catchment influences hydrograph and flood peak timing and magnitude. Methods, such as Geomorphic Instantaneous Unit Hydrograph (Rodriguez-Iturbe and Valdes, 1979), focus on predicting the travel time of water reaching channels and travelling downstream based the morphology of the catchment, drainage network and precipitation. Aquatic ecology takes a network-
55 centric approach, utilising dendritic ecological networks (Peterson et al., 2013). This method aims to take a spatially continuous view of rivers (Fausch et al., 2002) in order to appreciate the influence of flow and location in the network on discrete sites chosen for ecological sampling. Spatial statistical stream network models based on the branching of the network (Ver Hoef and Peterson, 2010) are shown to be more accurate than a standard Euclidean distance kriging model, yet only worthwhile if data sites are distributed across the network and are
60 spatially correlated (Peterson et al., 2013). Alternate methods for exploring relationships between network structure and ecological functioning are also based on Euclidean distance along the network (Ver Hoef and Peterson, 2010).

Each discipline represents the elements of the catchment critical to their field, focussing on describing catchment form, catchment flow responses and ecological responses. However, the geomorphology, hydrology
65 and ecology of the catchment are interconnected across spatial and temporal dimensions in the fluvial hydrosystem (Petts and Amoros, 1996). We argue that the overlap between disciplinary methods can be utilised to create a metric to represent the catchment that is meaningful across all disciplines and offers increased potential for effective catchment management utilising a multi-or interdisciplinary approach.

This paper develops integrated metrics, focussing on the topology of the river network as the key link between
70 the catchment and in-channel functioning. The metrics represent network density variation within catchments and have functional applications across the fields of geomorphology, hydrology and ecology (Sect. 1.1). The impact of internal network structure on physical habitat diversity patterns within catchments are explored by utilising datasets that are collected for regulatory purposes, with areas of higher network density likely to support higher habitat diversity (Sect. 1.2). The utility of the new topological metrics is compared against stream
75 order; a classic but over-simplified method of accounting for network topology. The new topology metrics are calculated for catchments with lower energy reaches and anthropogenic modification as much of the previous evidence for increases in diversity from high densities of links has been from highly-erosive mountainous catchments.

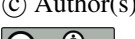


### 1.1 Network topology

River network structure, or network topology, is one way to conceptualise the integrated transport of water, sediment and nutrients from the upstream catchment to downstream reaches. The spatial arrangement of links (river channels) and nodes (confluences) concentrates the catchment effect in some areas of the landscape making network topology a useful architype of catchment functioning (Gupta and Mesa, 1988).

Drainage density (the total length of the network divided by catchment area) is most commonly used to compare
the amount of the catchment covered by river channels, but this fails to quantify spatial variation within catchments, and so offers only a partial means for functionally assessing catchment similarities and differences. To represent within catchment network structure stream order (Strahler, 1957), ordering river links along an up-to-downstream gradient based on their upstream connectivity (Fig. 1a), is also commonly used. However, stream order does not account for the spatial arrangement of links, only their relative position in the distance
dimension of the catchment. Conceptualising the catchment in this one-dimension leads also to over-simplification, for example, first order streams are often thought of as upland headwater streams, furthest away from the river mouth, yet often first order streams are tributaries to high-order, lowland streams with different characteristics than upland streams.

This paper argues that the spatial arrangement of links within catchments must be considered across the distance
*and* height of the catchment to obtain a full three-dimensional appreciation of catchment effect through network topology. Two methods from the field of hydrology - network width function (NWF; Kirkby, 1976) and link concentration function (LCF; Gupta et al., 1986) - offer increased dimensionality by accounting for the width of the network (i.e. the number of links) at successive distances, for the NWF or elevations, for the LCF.

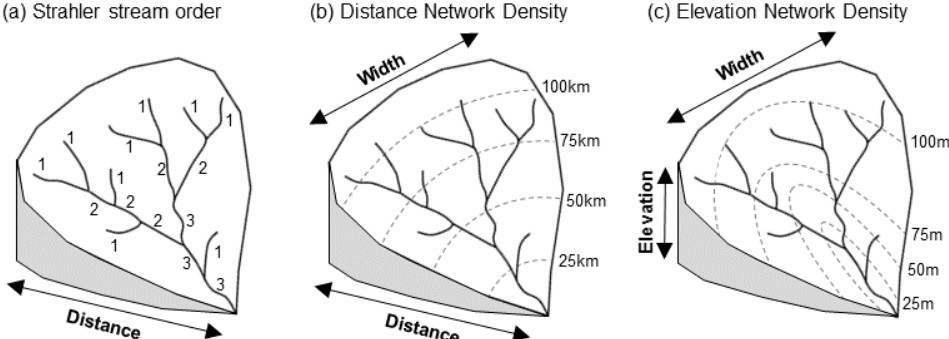

**Figure 1.** Topological metrics explored in this paper and the dimensions of the network they represent. **(a)** Strahler stream ordering representing only the distance dimension of the network. **(b)** Distance network density representing the width dimension of the network at each distance interval (inspired by the network width function (Kirkby, 1976)). **(c)** Elevation network density representing the width dimension of the network at each elevation interval (inspired by the link concentration function (Gupta et al., 1986)).



These methods quantify network topology within catchments with functional significance. NWF has hydrological application, representing the travel time of water through the network to predict the timing and magnitude of unit hydrographs and flood peaks (Rodriguez-Iturbe and Valdes, 1979) with a more functionally-specific method of than the traditional stream ordering approach (Gupta and Waymire, 1983). Extending applications beyond the field of hydrology, the timing and magnitude of the hydrograph has direct influence on instream ecology, controlling the formation, maintenance and disturbance of physical habitats (Bunn and Arthington, 2002). Longitudinal connectivity of water and sediment through the network is also one of the multiple dimensions of the fluvial hydrosystems approach to catchment ecohydrology (Petts & Amoros, 1996), influencing the capacity for lateral and vertical connectivity and the development of the riparian corridor over time. LCF is less frequently applied in hydrograph prediction than NWF. However, it may better reflect catchment hydrology by incorporating the effect of gradient on the travel time of water, rather than the constant travel time suggested by NWF (Gupta et al., 1986). These metrics also have morphometric significance, reflecting the internal shape of the network by segmenting catchments into intervals to represent how network density changes within catchments (Stepinski and Stepinski, 2005).

This paper adapts these methods to produce two new metrics: distance network density (adapted from the NWF) (Fig. 1b) and elevation network density (adapted from the LCF) (Fig. 1c). These metrics allow comparison and quantification of network topological variation both within and between catchment with improved interdisciplinary and functional applicability than the stream ordering approach.

**1.2 Network topology effects on in-channel functioning**

The topological structure of the river network configures the river ecosystem (Bravard and Gilvear, 1996) by impacting functioning at the reach and sub-reach scales. The distance dimension of the catchment, often represented by stream order (Fig. 1a), reflects up-to-downstream gradual changes exhibited by many in-channel features and species. It forms the basis of classic geomorphic models, highlighting the zones of sediment supply in the headwaters, sediment transfer in the mid-reaches and sediment storage near the outlet (Schumm, 1977). It is also a key component in classic ecological models such as the River Continuum Concept (Vannote et al., 1980) which describes gradual changes in grain-size, channel width, invertebrates, fish species and energy sources along the gradient. Both models suggest that diversity in in-channel morphology and biota may be highest in the mid-reaches as channels transition from erosional to depositional environments. The River Continuum Concept is a popular model but is critiqued for being too simplistic and for neglecting discontinuity introduced by changes at confluences (Perry & Schaeffer, 1987; Rice et al., 2001). Confluences, as nodes in the network, are associated with changes in hydrological, geomorphological (Best, 1987; Church & Kellerhals, 1978) and ecological (Kiffney et al., 2006; Rice et al., 2001) conditions and have therefore been termed biodiversity 'hotspots' (Benda et al., 2004b). Components of physical habitats important for instream biota, such as substrate size, flow type and in-channel morphology, become increasingly diverse because of hydrological and sedimentological changes at confluences. For example, substrate size changes from coarse to fine downstream along each "sedimentary link" in the network creating step-changes in sediment size, known as the Link Discontinuity Concept (Rice et al., 2001). Surface flow types, used to indicate the presence of physical biotopes (Rowntree, 1996), are also likely to become more diverse at confluences as the convergence of channels creates a number of different flow environments (Best, 1985) that result in different water-surface



topographies (Biron et al., 2002). The occurrence of channel features such as bars and boulders are also noted as impacts of confluences (Benda et al., 2004a).

The impact of confluences extends further than the immediate tributary junction, with channel diversity increased in the tributary and main channel upstream and downstream of the confluence (Rice, 2017). The Network Dynamics Hypothesis posits that catchments with higher drainage density, and thus more confluences, will have greater instream heterogeneity (Benda et al., 2004b), despite drainage density failing to be a useful catchment characteristic for predicting local habitat features (Davies et al., 2000). Not all confluences impact instream functioning (Rice, 1998) and studies suggest that confluences with similarly sized tributaries have the greatest impact on instream morphology (Benda et al., 2004a), the greatest flow diversity (Schindfessel et al., 2015), and greatest fish community diversity (Osborne and Wiley, 1992). The Network Dynamics Hypothesis suggests that catchment shape will influence the impact of confluences, as more compact catchments will have more similarly sized tributaries (Benda et al., 2004b) but in contrast, high densities of small tributaries flowing into a large channel have been suggested to cause small, cumulative changes within an area (Jones and Schmidt, 2016). These studies focus on the density of different sized confluences but alternate approach is that the position of confluences is key, for example, Milesi and Melo's (2013) study which concluded that distal regions of the catchment have more significant confluences.

Interestingly, there is little evidence of anthropogenic impacts at confluences in the literature but as confluences are concentration points of catchment effects it seems likely that they may be focal points for anthropogenic impacts. For example, flood events may occur downstream of large confluences as flood peaks converge creating the need for flood defence measures (Depettris et al., 2000). Also, sediment size at confluences is shown to increase in many studies (Church and Kellerhals, 1978; Knighton, 1980), but in tributaries whose watersheds are dominated by agricultural land uses, fine sediments may become dominant at confluences, potentially reducing habitat diversity (Owens et al., 2005).

## 2 Methods

### 2.1 Study sites

This study demonstrates the potential use of topological metrics for catchments with varying geologies and land uses in comparison to the highly-erosive environments considered by previous studies. The four selected catchments are from the Demonstration Test Catchment programme (Fig. 2) which are representative of 80% of soil and rainfall combinations in the United Kingdom (McGonigle et al., 2014).

The Avon and Wensum catchments have similar characteristics, both being dominated by chalk geology with lower average annual rainfall and a high percentage of arable farming land cover. In comparison, the Eden and Tamar are dominated by less permeable bedrock with higher average annual rainfall and a high percentage of grassland land covers. In terms of their morphometry, the Avon and Wensum both have an elongated shape and low drainage density. The Wensum is a low-relief catchment with a maximum elevation of 95 m. The Tamar has the smallest catchment area ($928 km^2$) and is the most circular. The Eden is the largest catchment ($2295 km^2$) with the highest maximum elevation (246 m).



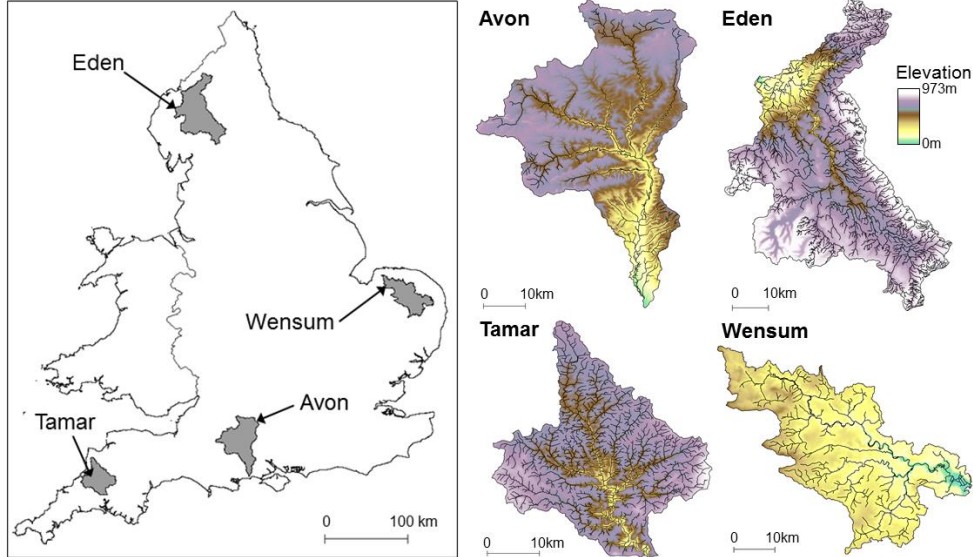

**Figure 2.** Demonstration Test Catchments. On the left, a map of catchment locations in England. On the right, topographic maps of each catchment with the river network shown in black.

## 2.2 Network topology metrics

Network topology metrics were calculated for each catchment using the 1:50,000 river network map, derived from both a Digital Terrain Model (DTM) and Ordnance Survey data (Moore et al., 1994). Anabranches and incorrectly digitised links in the network are removed using RivEX (Hornby, 2010). Removing anabranches was

necessary as the topological metrics were designed for dendritic networks so multi-thread channels, either naturally occurring or artificial ditches, would distort the calculations. This resulted in a total of 448, 2812, 1516 and 532 links in the Avon, Eden, Tamar and Wensum, respectively.

Elevation data was extracted from the Integrated Hydrological DTM (Morris & Flavin, 1994), a 50x50m gridded elevation raster with a 10cm vertical resolution. Average elevation of each link and the distance from

each link to the network outlet was extracted using RivEX (Hornby, 2010).

To extract a measure of network density that varies spatially within the catchment, each network is divided into 20 intervals, each of which in turn represent five percent of the total distance or highest elevation in the network. The network is divided in this manner based on the methods of the NWF and LCF which have functional application to hydrograph prediction. Twenty intervals provides a relatively coarse sampling of the network,

compared to the 100 intervals described by Stepinski and Stepinski (2005) when they adapted a morphometric variable, circularity ratio, to represent internal catchment elongation. Here, a total of twenty intervals is chosen so that most intervals contain links for the density calculation whilst ensuring the spatial distribution of network density within the catchment is characterised.





Distance network density was calculated following Eq. (1):

$$Distance\ network\ density = \frac{[n(d_0), \ldots, n(d_i), \ldots, n(d_N)]}{(d_N \times 0.05)} \tag{1}$$

where the number of links ($n()$) within each 5% distance interval ($d_i$) from the outlet ($d_0$) to the maximum distance in the network ($d_N$) normalised by the width of the interval ($d_N$ x 0.05).

Elevation network density was calculated following Eq. (2):

$$Elevation\ network\ density = \frac{[n(z_0), \ldots, n(z_i), \ldots, n(z_N)]}{(z_N \times 0.05)} \tag{2}$$

where the number of links ($n()$) within each 5% elevation interval ($z_i$) from the outlet ($z_0$) to the maximum height of the network ($z_N$) normalised by the width of the interval ($z_N$ x 0.05). Normalisation allows network densities to be compared between catchments controlling for differences in size and elevation as well as within catchments.

To assess the utility of the new multi-dimensional topology metrics in accounting for the spatial structure of the network, the metrics are compared to the one-dimensional Strahler stream order metric, extracted from the river network dataset using RivEX (Hornby, 2010).

### 2.3 Physical habitat diversity score

The impact of network topology on in-channel functioning is explored using a broad-scale approach, i.e. adapting data collected for regulatory compliance to answer scientific questions. Adapting such datasets to scientific enquiry allows analysis to be conducted in many catchments across a wide spatial extent. There are many habitat monitoring methods across the globe, with 121 survey methods recorded in over 26 different countries (Belletti et al., 2015), so this method may be adapted to other countries.

This study utilises the River Habitat Survey (RHS; Raven et al., 1996), a regulatory dataset collected by England's Environment Agency, which is used to reflect in-channel functioning in the catchments. This dataset has been used to identify catchment effects on habitats in broad-scale studies by previous research (e.g. Harvey et al., 2008; Naura et al., 2016; Vaughan et al., 2013) but none have included the effects of network topology.

Since 1994, over 24,000 sites have been sampled in catchments across England and Wales, including the Avon (n=418), Eden (n=398), Tamar (n=189) and Wensum (n=315). Surveys were conducted at random sites within each 10 km$^2$ of England and Wales to ensure geographic coverage, however this produces sampling bias as streams in high density areas will be under represented in the dataset, which is acknowledged in this study and discussed below.

At each site, over 100 features are recorded along a 500m reach with 10 "spot-check" surveys conducted every 50m and a "sweep-up" survey conducted across the whole reach (see Raven et al. (1996) for details). An individual score is assigned to each component of the survey based on the diversity of features recorded (Table 1).





**Table 1.** RHS components used in the physical habitat diversity score. Each component reflects physical habitat properties. Individual component scores are assigned based on dominant features observed at 10 spot-checks along the 500m reach and reflect the diversity of the component within the reach.

| RHS component | Physical habitat properties | Features recorded | Component score calculation (Raven et al., 1998) |
|---|---|---|---|
| *Flow type* | Flow types represent biotopes which reflect the habitat of the entire biological community (Udvardy, 1959). | Free-fall, chute, broken standing waves, unbroken standing waves, chaotic, rippled, upwelling, smooth, no perceptible flow | ***Score 1*** for each flow type recorded. ***Score 2*** if it occurs at 2-3 spot-checks. ***Score 3*** if it occurs at 4+ spot-checks *or* if only one type occurs at all 10 spot-checks ***Score 0*** if channel is dry. (Additional scores available from sweep-up survey) |
| *Substrate size* | Substrates of different sizes provide a functional habitat for different species (Harper et al., 1992) | Bedrock, boulder, cobble, gravel/pebble, sand, silt, clay, peat | ***Score 1*** for each substrate type recorded *or* if substrate is not visible in 6+ spot-checks ***Score 2*** if it occurs at 2-3 spot-checks. ***Score 3*** if it occurs at 4+ spot-checks *or* if only one type occurs at all 10 spot-checks |
| *Channel features* | Erosional and depositional features reflect the stability of the channel (Bizzi and Lerner, 2015) | Exposed bedrock/boulders, unvegetated mid-channel bar, vegetated mid-channel bar, mature island | ***Score 1*** for each channel feature recorded. ***Score 2*** if it occurs at 2-3 spot-checks. ***Score 3*** if it occurs at 4+ spot-checks (Additional scores available from sweep-up survey) |

The individual scores, derived by expert opinion, combine to form a Habitat Quality Assessment score which is used to determine the diversity and naturalness of the riparian zone at each site in accordance to regulatory policy. This study only considers in-channel responses to network topology and calculates a physical habitat diversity score for each site following Eq. (3):

$$Physical\ habitat\ diversity\ score = Flow\ type\ score + Substrate\ score + Channel\ features\ score \quad (3)$$

where individual scores of flow type, substrate size and channel feature diversity (Table 1) are totalled.

For each distance and elevation interval created by the network topology metrics, the mean, maximum and minimum physical habitat diversity score was calculated. Despite the sampling bias towards less dense areas of the network, most distance and elevation intervals contained RHS sites (with only some low density intervals not containing RHS sites). For distance or elevation intervals with less than three RHS sites, maximum and
minimum values were removed. This method is designed to account for natural variation and modification at individual RHS sites, in order to assess broad patterns of habitat diversity at the catchment level.





### 2.4  Statistical analysis

Analysis was conducted with all catchments combined into a single population to identify overall trends across all catchments, a method used in previous broad-scale studies. The analysis was also split into individual
catchments to identify how the relationship between network topology metrics and physical habitat diversity differed between catchments.

#### 2.4.1 Spearman's Rho Correlations

Correlation tests were used to determine the strength and direction of the association between mean, maximum and minimum habitat variables and distance network density and elevation network density to ascertain how
habitat diversity responds to network topology. Spearman's Rho correlation method was used as the datasets have non-normal distributions.

As multiple correlations are conducted, False Discovery Rate (Benjamini and Hochberg, 1995) corrections were applied to the p-values produced from the Spearman's Rho correlations to reduce the risk of type I error. The False Discovery Rate method has been found to be more powerful than other procedures for controlling for
multiple tests (Glickman et al., 2014).

#### 2.4.2 Mann-Whitney U tests

The correlations with the new network topology metrics are compared to habitat diversity variation between stream orders to identify whether the new topological metrics of network density examined here build on explanations of habitat diversity patterns by stream order. Pairwise Mann-Whitney U tests with False Discovery
Rate corrections to the p-value threshold were conducted to identify which stream orders had distributions of habitat diversity that were statistically different from others.

### 3    Results

#### 3.1 Differences in network topology metrics between catchments

The topological metrics developed in this study show the internal structure of the network for each catchment.
The separation of the catchments into distance and elevation intervals highlight different features of the catchment. The distance intervals (Fig. 3a) are arranged longitudinally within the catchment, highlighting sub-basins within each catchment. The elevation intervals (Fig. 3b) have a radial arrangement, centring around the incised main channel of each catchment.

Distance network density (Fig. 3c) reflects the higher numbers of links in the Eden and Tamar compared to the
Avon and Wensum. However, interestingly the highest distance network density is recorded in the Tamar, the smallest catchment by area. The shape of the distance network density function reflects the internal shape of the network. For example, the Tamar has a peaked density distribution reflecting the circular shape of the catchment such that the majority of links are at 55%-65% distance from the catchment outlet. The Avon and Eden reflect similar internal network structures, both exhibiting a bimodal density distribution, despite the differences in the
number of links in the catchments. The density distribution of the Wensum has a more complex internal distribution of links with multiple peaks in density.







**Figure 3.** Network topology metrics for each catchment. **(a)** Percentage distance from the outlet intervals used to calculate distance network density. **(b)** Percentage elevation intervals used to calculate elevation network density. **(c)** Distance network density metric. **(d)** Elevation network density metric. **(e)** Number of links classified as each Strahler stream order.





Elevation network density (Fig. 3d) shows similar density distribution shapes to distance network density, with a unimodal distribution for the Tamar and multi-modal distributions in the other catchments. In contrast to distance network density, elevation network density shows the highest peaks in density in the Wensum, the catchment with lowest network elevation. The peak densities in the Wensum occur at similar positions in the elevation and distance intervals, whereas, the peaks in the other catchments are negatively skewed, showing the network density is highest at low-mid elevations.

The percentage of links classified as each stream order exhibit the same pattern across all catchments (Fig. 3e). Nearly half of the links in each catchment are classified as first order streams and the number of links declines exponentially towards the highest orders.

### 3.2 Physical habitat diversity relations with network topology metrics

Mean physical habitat diversity scores are higher in the Eden (mean=19) and Tamar (mean=20), than the Avon (mean=12) and Wensum (mean=11). When all catchments are combined mean, maximum and minimum physical habitat diversity show significant positive correlations with the distance network density topology metric (Fig. 4a). However, significant correlations were not consistent when analysis was split into individual catchments. The Eden and Tamar both show positive correlations across maximum, mean and minimum habitat diversity. This indicates that increases in network density cause an overall amplification of habitat diversity scores in each distance interval (however, positive correlations with mean diversity in the Tamar and minimum diversity in the Eden and Tamar were not significant). In contrast, the Avon and Wensum show positive correlations with maximum and mean habitat diversity but negative correlations with minimum diversity. In these cases, sites with the highest diversity become more heterogeneous in distance intervals with high network density, but the least diverse sites in each interval become more homogenous.

However, correlations with minimum physical habitat diversity are not significant in the Avon once the False Discovery Rate correction was applied and the Wensum has no significant correlations suggesting that network density has little influence on the catchment's physical habitat diversity.

Elevation network density shows no significant correlations with habitat diversity when all catchments were combined or in individual catchments (Fig. 4b). The elevation network density correlations show that maximum habitat diversity increases with network density in individual catchments, as with distance network density.

The range of physical habitat diversity is broad across the majority stream orders in all catchments (Fig. 4c). When all catchments are combined, physical habitat diversity score peaks in the mid-reaches with $2^{nd}$, $3^{rd}$ and $4^{th}$ order streams being significantly different (Mann Whitney U test – $p<0.05$) from $5^{th}$ and $6^{th}$ order streams. In each individual catchment, the largest stream order has a significantly lower median habitat diversity than other stream orders. Aside from this, the relationship between physical habitat diversity scores and stream order varies between catchments. Peak median diversity occurs in $2^{nd}$ order streams in the Avon, but in $3^{rd}$ and $4^{th}$ order streams in the Eden. Ranges of physical habitat diversity are relatively consistent in the Tamar and Wensum except for the significant decline in diversity in $6^{th}$ order streams.





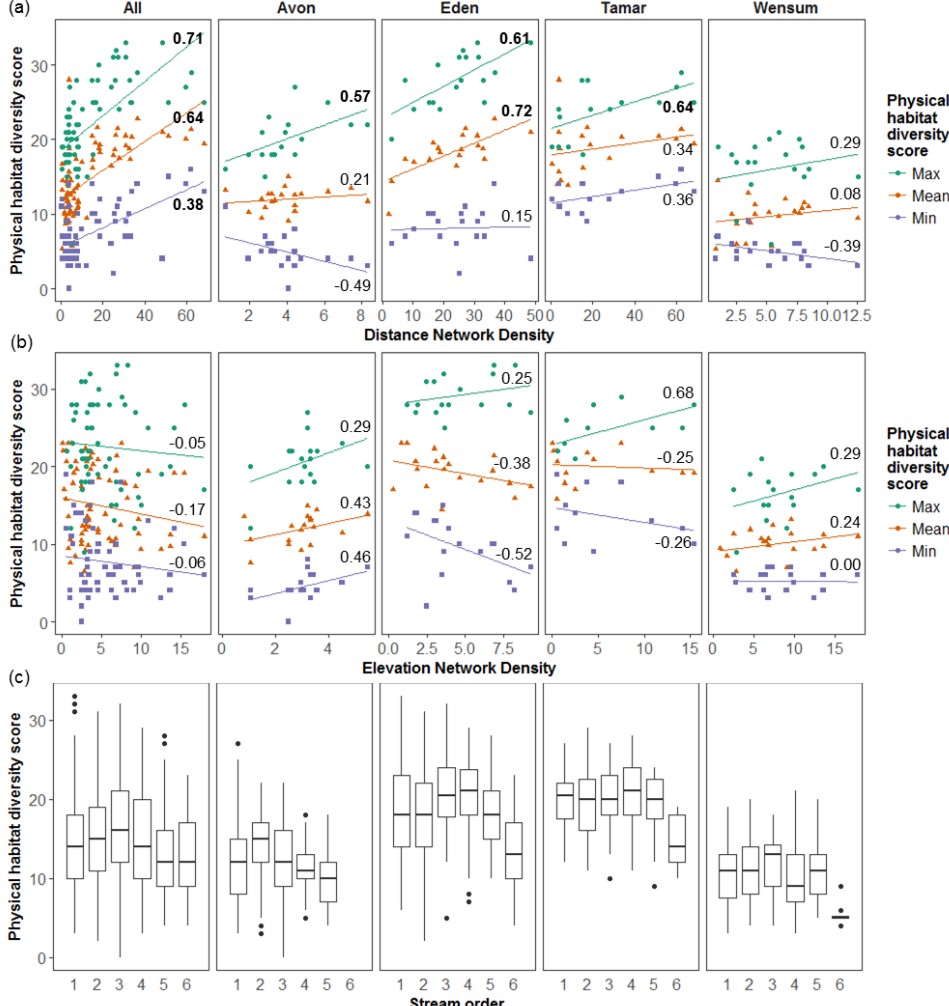

**Figure 4.** Relationships between network topology metrics and physical habitat diversity score in All catchments combined and each individual catchment. **(a)** Spearman's Rho correlations between mean, maximum and minimum habitat diversity and Distance network density. **(b)** Spearman's Rho correlations between mean, maximum and minimum habitat diversity and Elevation network density. Significant correlation coefficients ($p < 0.05$) shown in bold. **(c)** Physical habitat diversity score distributions for stream orders.





## 4 Discussion

### 4.1 A new approach to utilising network topology in catchment-level analysis

Network density metrics represent an alternative approach to account for network topology in catchment-level
studies, optimising the width dimension of the network (or the number of links in the network) as opposed to the
commonplace stream order metric which only reflects the longitudinal position of links in a network (**Figure 1**).
This study shows that two new topology metrics can be calculated simply from a DTM with GIS and, using a
broad-scale analysis of river attributes, can be used to investigate the functional processes within catchments.

Elevation has been a key metric in explaining observations of RHS variables including flow type, substrate, etc.
in a number of studies (Jeffers, 1998; Naura et al., 2016; Vaughan et al., 2013), so it is unsurprising that more
streams, at a greater range of elevations, will introduce greater habitat diversity than intervals with more streams
at the same elevation.

The spatial configuration of the distance and elevation intervals used in the calculation of network density may
also impact the effectiveness of each topological metric. Distance network density separates the catchment into
intervals based on distance which radiate from the outlet (Fig. 3a), reflecting natural sub-basins within the
fractal structure of the catchment (Lashermes and Foufoula-Georgiou, 2007). This differs from elevation
network density which separates the catchment into intervals based on elevation, thus forming contours
radiating from the main channel of each catchment network (Fig. 3b). The configuration means that distance
intervals contain streams that are in closer proximity to one another rather than the more distal configuration
created by the elevation intervals, suggesting a degree of spatial dependency in river functioning. This has been
highlighted in previous studies where spatial network structure has a stronger influence on some in-channel
processes than predictor variables such as elevation (Steel et al., 2016).

Distance network density, which accounts for the width of the network along the distance dimension of the
catchment, is the more successful metric with more significant and stronger correlations with physical habitat
diversity than the elevation network density metric (Fig. 4a and b). This may be because each distance interval
contains a broader range of elevations than elevation intervals within which elevation range is controlled.
Impacts of network topology on instream physical habitats

Higher network density was expected to be related to higher levels of habitat diversity due to previous research
stating that confluences in the network increase diversity and therefore catchments with higher drainage density
will have higher morphological heterogeneity (Benda et al., 2004b). The results of the correlations between
distance network density and physical habitat diversity score support this assumption (Fig. 4a). For example,
when catchments are grouped together the results show that habitat diversity increases in network dense distance
intervals, where there are more confluences.

The results from individual catchments give further insight into the degree of influence that distance network
density has on habitat diversity (Fig. 4a). The maximum habitat diversity within distance intervals significantly
increases with network density in all but one catchment, the Wensum. However, only the Eden shows a positive
relationship between mean habitat diversity and network density, implying that in other catchments, greater
network density only increases diversity at some sites but not enough to significantly impact mean diversity.





This may reflect findings from previous studies which suggest that not all confluences cause reach-scale
changes (Rice, 1998) and that catchment shape influences the amount of significant confluences, with linear
catchments containing approximately half as many significant confluences than compact catchments (Rice,
2017). The two catchments with the strongest correlations between distance network density and maximum
habitat diversity, the Eden and Tamar, are the most compact catchments and the Wensum is the most linear
suggesting that catchment shape may be influencing the effect of network density on habitat diversity. Further
analysis on a greater range of catchment shapes is needed to fully support this conjecture.

There is evidence that in the Avon and Wensum, increasing network density reduces minimum habitat diversity,
suggesting that network density both introduces increases heterogeneity and homogeneity within distance
intervals. This may be because these catchments have a greater percentage of arable farmland than the other
catchments so may experience greater inputs of fine sediments, reducing substrate size diversity. Other
anthropogenic activities in response to concentrations of fine sediment, such as dredging may limit the amount
of natural in-channel features and flow types. However, it must be noted that none of the negative correlations
with minimum habitat diversity were significant once the False Discovery Rate correction had been applied.

The lack of significant results for the Wensum, and mean and minimum diversity in other catchments, suggest
that low habitat diversity scores are frequently influenced by external factors such as anthropogenic
modifications which are independent of network density. This suggests that the Network Dynamic Hypothesis
theory that higher drainage density (a catchment-scale metric of topology) equates to higher morphological
heterogeneity (Benda et al., 2004b) is too simplistic. This is because network density varies spatially within
catchments, creating spatial variation in habitat diversity scores, and also that external factors influence physical
habitats in catchments.

**4.3 Comparison of stream order to new topological metrics**

The classic method of accounting for network topology, stream order, is critiqued for failing to represent
discontinuities in the network and simplifying the network into a gradient. The number of links in different
stream orders is consistent across all catchments (Fig. 3e) not reflecting the internal structure of the network or
the variety between catchments that is achieved by the distance network density and elevation network density
metrics. The analysis of the two new metrics presented in this paper shows that distance from source and
elevation are not mutually exclusive, contrary to the stream order metric which represents streams as upstream
to downstream or upland to lowland.

Stream order does not sufficiently represent the diversity introduced to the channel by the complexity of the
network. In some cases, there was evidence of increased median diversity in mid-reaches which is suggested by
both geomorphic (Schumm, 1977) and ecological frameworks (Vannote et al., 1980). However, the ranges in
physical habitat diversity scores were broad across all stream orders, with the most significant finding being that
the highest order stream orders in catchments were generally the least diverse (Fig. 3e).

This study has found little reason to suggest that the purely longitudinal stream order metric is effective for
explaining patterns of habitat diversity in river networks. Others have also found that stream order has weak and
inconsistent relationships with biodiversity patterns in river systems, providing insufficient explanation of the





mechanisms controlling biodiversity (Vander Vorste et al., 2017). Instead, this study finds that distance network density is a more powerful metric which conceptually provides an improved method of accounting for the impacts of network topology on the fluvial system exhibiting strong relationships with habitat diversity, particularly maximum habitat diversity (Fig. 4a).

**4.4 Applicability of network topology metrics to different environments**

Much of the seminal work on network and confluence impacts (e.g. the Network Dynamic Hypothesis; Benda et al., 2004b, and Link Discontinuity Concept; Rice et al., 2001) is conducted in natural, highly erosive catchments with first hand empirical measurements. However, in an age when rivers and their catchments are increasingly altered by anthropogenic modification (Meybeck, 2003), contemporary studies must not only aim to expand knowledge but find methods of transferring knowledge to many, increasingly altered, catchments (Clifford, 2002).

The catchments selected by this study are more greatly modified and, although they reflect a range of fluvial environments in England, are more representative of lower energy catchments than the seminal studies. Benda et al. (2004a) suggests that confluence effects in less active landscapes would be subdued compared to highly erosive landscapes. The evidence presented here demonstrates the utility of evaluating network topological structure in studies on catchment-level effects in any type of fluvial environment because of the continuing relevance of network topology regardless of the low energy and increasingly widespread anthropogenic changes in catchments. However, it must be noted that the catchment with the lowest elevation range, the Wensum, showed no significant results with distance network density or elevation network density (Fig. 4a and b). This suggests that these topological metrics may only have a functional effect on in-channel diversity with sufficient energy, a topic which should be explored further in future research.

The methods presented in this paper are designed to be implemented in any catchment with a dendritic network structure. The topology metrics can easily be calculated from any dendritic network with DTM data using GIS and compared to any site scale data. This study uses regulatory monitoring datasets so that analysis is targeted to in-channel features of interest to river managers. Also, the high volume and wide spatial extent of data available from regulatory sources allows for between catchment comparisons. The influence of network topology on in-channel functioning is likely to vary between catchments depending on the modification and energy of the fluvial system.

**5 Conclusions**

Although appreciation of catchment-level effects is considered the epitome of understanding river functioning, the key component of the catchment - the river network - is overlooked and oversimplified by catchment-level studies. This study finds that river network topology plays a role in structuring the distribution physical habitat diversity throughout the catchment, offering more detailed explanation than the classic stream order metric. Network dense areas are generally found to have greater habitat diversity and in some cases where agricultural land use is more dominant, a potentially negative impact on habitat diversity. The study also highlights that anthropogenic modification and other factors mean that network density only has an impact at some sites in the catchment. This paper shows that network topology itself may be in-part influenced by catchment





characteristics, demonstrating that the inclusion of network topology in catchment-level studies adds a layer of function-based understanding to such studies.

The broad-scale methodology adopted by this study allows the most successful network topology metric, distance network density (which is easily be extracted from open-source data using GIS software), to be compared to any regulatory dataset. The use of regulatory datasets not only allows for analysis over a wider spatial extent but also for more applicable results for regulatory bodies. Therefore, the interdisciplinary approach to characterising network topology can be applied efficiently and effectively to capture catchment-

level impacts on in-channel functioning in any catchment across the globe.

**Appendix A**

**Table A1.** Significant correlations (p<0.05*) between physical habitat diversity scores and **(a)** Distance network density and **(b)** Elevation network density for all catchments combined and each catchment. Table also shows p-values with the False Discovery Rate (FDR) correction to account for the number of correlations conducted.

(a)

| | All catchments | | Avon | | Eden | | Tamar | | Wensum | |
|---|---|---|---|---|---|---|---|---|---|---|
| | p-value | FDR p-value | p-value | FDR p-value | p-value | FDR p-value | p-value | FDR p-value | p-value | FDR p-value |
| *Mean diversity* | <0.01* | <0.01* | 0.39 | 0.45 | <0.01* | <0.01* | 0.17 | 0.25 | 0.21 | 0.26 |
| *Max diversity* | <0.01* | <0.01* | 0.01* | 0.02* | 0.01* | 0.02* | 0.01* | 0.02* | 0.74 | 0.74 |
| *Min diversity* | <0.01* | 0.01* | 0.03* | 0.06 | 0.56 | 0.60 | 0.19 | 0.26 | 0.10 | 0.17 |

(b)

| | All catchments | | Avon | | Eden | | Tamar | | Wensum | |
|---|---|---|---|---|---|---|---|---|---|---|
| | p-value | FDR p-value | p-value | FDR p-value | p-value | FDR p-value | p-value | FDR p-value | p-value | FDR p-value |
| *Mean diversity* | 0.15 | 0.38 | 0.08 | 0.30 | 0.10 | 0.30 | 0.38 | 0.52 | 0.30 | 0.50 |
| *Max diversity* | 0.70 | 0.75 | 0.26 | 0.50 | 0.33 | 0.50 | 0.02* | 0.23 | 0.27 | 0.50 |
| *Min diversity* | 0.65 | 0.75 | 0.06 | 0.30 | 0.03* | 0.23 | 0.44 | 0.55 | 0.99 | 0.99 |







**Table A2.** Significant differences (p<0.05*) in physical habitat diversity scores between stream orders for all catchments combined and each catchment using pairwise Mann-Whitney U tests with False Discovery Rate (FDR) correction. (n/a – the Avon has no 6th order streams).

| | | | All catchments | Avon | Eden | Tamar | Wensum |
|---|---|---|---|---|---|---|---|
| | | | FDR p-value | FDR p-value | FDR p-value | FDR p-value | FDR p-value |
| 1st | - | 2nd order | 0.04* | <0.01* | 0.76 | 0.99 | 0.48 |
| 1st | - | 3rd order | <0.01* | 0.71 | 0.01* | 0.91 | 0.06 |
| 1st | - | 4th order | 0.16 | 0.81 | 0.02* | 0.49 | 0.72 |
| 1st | - | 5th order | 0.06 | 0.02* | 0.99 | 0.99 | 0.53 |
| 1st | - | 6th order | 0.11 | n/a | <0.01* | 0.05 | <0.01* |
| 2nd | - | 3rd order | 0.11 | <0.01* | <0.01* | 0.91 | 0.29 |
| 2nd | - | 4th order | 0.61 | <0.01* | 0.01* | 0.49 | 0.27 |
| 2nd | - | 5th order | <0.01* | <0.01* | 0.78 | 0.99 | 0.82 |
| 2nd | - | 6th order | <0.01* | n/a | <0.01* | 0.07 | <0.01* |
| 3rd | - | 4th order | 0.08 | 0.71 | 0.84 | 0.49 | 0.02* |
| 3rd | - | 5th order | <0.01* | 0.01* | 0.02* | 0.86 | 0.08 |
| 3rd | - | 6th order | <0.01* | n/a | <0.01* | 0.02* | <0.01* |
| 4th | - | 5th order | <0.01* | 0.01* | 0.02* | 0.46 | 0.16 |
| 4th | - | 6th order | 0.02* | n/a | <0.01* | 0.02* | <0.01* |
| 5th | - | 6th order | 0.69 | n/a | <0.01* | 0.07 | <0.01* |

*Competing interests.* The authors declare that they have no conflict of interest.

**Acknowledgements**

The authors would like to thank the Environment Agency and the Centre for Ecology and Hydrology for access to the data used in this paper. The data used are listed in the references. This research is funded by the Natural Environmental Research Council (NERC).

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
