# Peer review of "Integrating network topology metrics into studies of catchment-level effects on river characteristics"

_Hydrology and Earth System Sciences, 2018_

## Referee Comment (RC1) · S. Bizzi (Referee) · 1 Jun 2018

The paper "Integrating network topology metrics into studies of catchment-level effects on habitat diversity" develops new metrics of network topology to predict diversity of in-channel morphological habitat observed in different river basins in UK. The paper's intent is meaningful because current availability of RS data (such as high resolution DEM nowadays widely spread worldwide) allow to calculate river network metrics capable to include more information on topographical and topological spatial organization of fluvial systems compared to more traditional network metrics such as the Strahler order (or Shreve order). The new metrics proposed in this paper can be calculated

easily in most of the basin worldwide and this is an important credit of the work which foster future applicability. The main limitation I found in the work is the use of the River Habitat Survey as metrics to assess habitat diversity (see specific comments for more details on this point). On the other hand I do admit that its wide application in UK at national scale over the years make of this database a valuable and unique source of information for this type of analysis and its replicability. The paper findings though show a moderate and sometime weak correlations between the new network topology metrics and habitat diversity. The authors do a good job in explaining possible causes and commenting different source of uncertainty. I also find reference to literature well developed and the statistical analysis scientifically sound. However, I think the paper needs some major revision about the description of the network topology indexes, which is little confusing at this stage. Moreover, the limitation of the current work regarding the use of RHS should be more extensively debated. If these limitations will be addressed I'm happy to suggest the paper for publication since I believe the topic is of interest to the reader of HESS and it has wide opportunities of future applications.

MAJOR COMMENTS:

- Sections 1.1 and 1.2 should be merged together. Many concepts present in Sect1.1 are repeated in Sect1.2 and I don't find a clear division between introduction to "Network topology" in Sect. 1.1 and its "effects on in channel functioning" in Sect 1.2 as suggested by their title. Sect 1.2 focuses extensively on confluences but the reader yet do not understand why they are so central in the network topology metrics and you need to clarify if the metrics proposed describe primarily how confluences are spatially organized or also other network features. - Section 2.1 start with few sentences (lines 170-175) which sound more as paper objectives and for this reason should be moved into the introduction. Here, I suggest to revise lines 70-78 to clarify paper objectives. - My main concern about this work is about the use of the RHS as habitat diversity evidences. The scoring system adopted is not robust. It simply evaluates 'as better' more frequent presence of specific features. This is a very subjective criterion, which cannot be linked easily to any geomorphological processes associable to various network topologies. Because this is the main physical hypothesis at the basis of your analysis this limitation need to be more extensively debated in the paper. Ideally you should encourage the adoption of more processes based river channel classifications, where morphological forms are classified based on different channel processes and then can be associated to different network topologies on the basis of physical principles (here some literature as reference, Gurnell, A. M., M Rinaldi, B Belletti, S. Bizzi, B Blamauer, G Braca, T Buijse, et al. ÂńA multi-scale hierarchical framework for developing understanding of river behaviour to support river managementÂż. Aquatic Sciences 78(1) (2016): 1–16. https://doi.org/10.1007/s00027-015-0424-5. Brierley, G J, e K A Fryirs. Geomorphology and River Management: Applications of the River Styles Framework. Oxford: Blackwell Science Ltd, 2005). The discussion sections should dedicate a paragraph to address these limitations and propose future applications capable of balancing the good availability of RHS data (in case of UK) with the need of better classification schemes. - In the result sections you often comment how the presence of anthropic pressures may alter the statistical correlations analysed. This is a good point. However, RHS database does includes quite an elevate number of features regarding in-channel artificial features. It would be very interesting to see if an indicator of artificial channel degree extracted from the RHS DB is significantly correlated with the unexplained variance of your model. This would be a notable finding, and also a good use of the RHS variables. - RHS is sampled on reach of 500 meter in any river (and it is not suitable for large rivers). For this reason the frequency of features used in your scoring systems of habitat diversity is likely deeply affected by river channel size (bigger river have larger river channel features). This limitation should be commented. You should try to see if there is a bias in your statistical analysis associated with basin area. - The paper points out more times that your proposed network topological metrics are more informative than Strahler order ( see for instance lines 411-414). This is very likely true, based on type of information included in these indexes. However, I would appreciate a wider prospective commenting the indexes. For instance I don't think they are

mutual exclusive as sometime it seems from your judgments. They provide measures of different basin features: for instance Strahler provides information on the longitudinal development of river networks, whereas your indexes describe how confluences density are organizer from distance from the outlet on based on topography. Would be nice to see how the ability to predict habitat diversity (though the limitation of the metric used) improve using all the three indexes, i.e., a multivariate regression. This may help to explain if the indexes are mutual exclusive, or if each of them explain different aspects of habitat diversity. This may help to disclose the physical link between network topological features and habitat diversity. In case you don't want to develop this additional analysis, you should justify in the discussion why you don't think this is necessary and debate this issue.

MINOR

- Lines 141: "For example, substrate size changes from coarse to fine downstream along each "sedimentary link"". Yes but sometime also fine to coarse. Please revise. - Lines 159-161 are unclear to me, please revise. - Lines 353-357 are unclear to me, please revise. - Lines 360-361 "This may be because each distance interval contains a broader range of elevations than elevation intervals within which elevation range is controlled. Impacts of network topology on instream physical habitats". Very unclear to me, are you saying that elevation is more important because it varies more in the 'distance' metric ? If true, this would be a strong statement which affect your results and the meaning of your indexes and should be much more extensively debated. - Lines 428-431: lack of correlation does not depend only on channel energy but also on network forms, indeed Wensum has the most "linear" shape. Can you better explain this point ? - Lines 445-447 ("The study also highlights that anthropogenic modification and other factors mean that network density only has an impact at some sites in the catchment".). You cannot state that. Your findings did not prove that, since you have no quantitative information on the level of anthropic pressures in your sites ( see my suggestion to add this information from the RHS database). You can presume it based

on previous literature. Please revise this point. - Lines 447-448 ("This paper shows that network topology itself may be in-part influenced by catchment characteristics. . ."). When ? How? It is unclear to me. Please explain. - Appendix A is never cited in the main text. - Caption of Figure 4: "Significant correlation coefficients (p<0.05) shown in bold" should refer to (a) and not (b).

————————————————

---

## Referee Comment (RC2) · Anonymous Referee #2 · 10 Mar 2019

The authors try to quantify the effect of network topological metrics on habitat diversity using examples of four diverse catchments in the UK. The idea is interesting, and the paper is written well. However, there are some major issues in the paper that need to be addressed before publication. Thus, I would recommend major revisions, and my main comment related to bullet 2. Following are my major and minor comments:

1. The authors claim in the abstract, and then in the Introduction, that they have developed two new network metrics – however, I struggle to see how the metrics are new. The distance network density is the same as the width function, while the elevation network density is the same as the link concentration factor. The authors say that they

have adapted these metrics, but all I see is that the authors have used the metrics. Adaptation of the metrics would involve changing them in some specific way, and this has not been done to my understanding. If this is not correct, the authors need to clarify. However, if it is correct then the authors need to be clear that the development of the metrics is not the contribution of their research..it is rather the application of the metrics. 2. The authors state in the abstract "The results indicate that the new metrics offer a richer, and functionally more-relevant description of network topology than stream order, highlighting differences in the density and spatial arrangement of each catchment's internal network structure". However, when I read the paper I struggle to see that the evidence really points to this statement. My understanding is that this statement is based on Figure 4. However, figure 4 does not really show this effectively. The authors do correlations with the network topology metrics, but there is no correlation attempted with stream order, which make it a difficult comparison to make. Moreover, the correlations with the new metrics are weak and most of them are non-significant. This makes it an extremely difficult argument to make. The novelty of this paper is that these new metrics are better – however the evidence that the authors present does not convince me of this. 3. One suggestion for figure 4 might be to attempt to correlate median and 1st and 3rd quartiles, instead of mean, max and min. Max and min values can be highly erratic with environmental data, and might not always be amenable to such analysis 4. Line 410 – this is extremely qualitative – this section should focus on the authors findings, but instead becomes more of a lit review..and no substantive reason to argue for a lack of reason. Is there a p value and r2 for one set of correlations that is better than the other? 5. Line 380 – why does increasing network density lead to reducing minimum habitat diversity but increasing max habitat diversity? The arable land explanation provided here is not clear The metrics should be described in the abstract to make the statements made in the abstract clearer.

---

## Author Comment (AC1) · 3 Apr 2019

Thank you very much for taking the time to read our manuscript and for your constructive comments.

You have highlighted some key limitations which had not been properly addressed in the original manuscript (i.e. issues surrounding the River Habitat Survey data, the Habitat Quality Score and the mutual exclusivity of the metrics) and identify areas where analysis may be improved upon (i.e. your suggestion to include Habitat Modification Score and a multi-variate regression analysis) in addition to other comments.

Based on your comments, we have revised the manuscript (see Supplement) and provide a table (see Supplement) detailing our specific response to each of your concerns, along with the concerns of our 2nd referee. On occasion where we dispute part of your comment so that it is not fully rectified in the revised manuscript, it is indicated in the table with the phrase "Part-addressed". We have added new variables from the River Habitat Survey to our analysis and so the figures, results and discussion have been updated accordingly.

We believe the revised manuscript now adds more to the discussion surrounding the impacts of network topology on river reach characteristics and is of improved scientific rigor. We hope you share our view and that the changes we have made satisfy your concerns with the original manuscript.

Please also note the supplement to this comment:
https://www.hydrol-earth-syst-sci-discuss.net/hess-2018-89/hess-2018-89-AC1-supplement.zip

---

## Author Comment (AC2) · 3 Apr 2019

Thank you very much for taking the time to read our manuscript and for your constructive comments.

You have highlighted some key concerns regarding the boldness of some claims made in the original manuscript (i.e. the novelty of the network metrics and our conclusions based on mid-weak strength correlations) and identify areas where the statistical analysis employed may be improved upon (i.e. your suggestion to use correlation for the stream order variable and to use median, 1st and 3rd quartile values) in addition to other comments.

[Figure]

Based on your comments, we have revised the manuscript (see Supplement) and provide a table (see Supplement) detailing our specific response to each of your concerns, along with the concerns of our 1st referee. On occasion where we dispute part of your comment so that it is not fully rectified in the revised manuscript, it is indicated in the table with the phrase "Part-addressed". We have altered components of our statistical analysis at your suggestion and so the figures, results and discussion have been updated accordingly.

We believe the revised manuscript now adds more to the discussion surrounding the impacts of network topology on river reach characteristics and is of improved scientific rigor. We hope you share our view and that the changes we have made satisfy your concerns with the original manuscript.

Please also note the supplement to this comment:
https://www.hydrol-earth-syst-sci-discuss.net/hess-2018-89/hess-2018-89-AC2-supplement.zip
* * *